# Paramacular Choriocapillaris Atrophy

**DOI:** 10.3390/biomedicines11072074

**Published:** 2023-07-24

**Authors:** Ivona Bućan, Kajo Bućan

**Affiliations:** 1Eye Clinic, University Hospital Centre Split, 21000 Split, Croatia; kbucan@kbsplit.hr; 2Department of Opthalmology, School of Medicine, University of Split, 21000 Split, Croatia

**Keywords:** foveal sparing, choriocapillaris atrophy, macular atrophy, retinal dystrophy, choroidal dystrophy

## Abstract

In this paper, a review of a rare case of paramacular choriocapillaris atrophy with a foveal-sparing phenotype is carried out. The 73-year-old patient stated that they had impaired vision and photophobia in both eyes during a regular ophthalmological examination, denying visual field defects and night blindness. A complete ophthalmological examination (best-corrected visual acuity, applanation tonometry, and biomicroscopy of anterior and posterior segments) and diagnostic tests, including fundus autofluorescence, fluorescein angiography, optical coherence tomography with angiography, computerized perimetry, and electroretinography, were carried out. The underlying genetic pattern is unclear, which points to paramacular choriocapillaris atrophy. According to recent research on histology, pathologies categorized as regional choroidal dystrophies are caused by alterations at the level of the retinal pigment epithelium. Despite the unresolved etiopathogenetic mechanism of foveal sparing in central choroidal and retinal dystrophies, a highly variable disease phenotype with spared fovea and central visual acuity present in a variety of heterogeneous dystrophies supports a disease-independent mechanism that allows the survival of foveal cones. The related preservation of BCVA has implications for individual prognosis and influences how treatment trials for choroidal and retinal dystrophies are designed.

## 1. Introduction

According to recent histology research, pathologies that are categorized as choroidal dystrophies and Stargardt disease are caused by alterations at the level of the retinal pigment epithelium (RPE) [1]. Despite the unresolved etiopathogenetic mechanism of foveal sparing in central choroidal and retinal dystrophies, a highly variable disease phenotype with spared fovea and central visual acuity present in a variety of heterogeneous dystrophies supports a disease-independent mechanism that allows for the survival of foveal cones. Patients over 50 years old are more likely to experience choroidal and retinal dystrophies of the posterior pole with foveal sparing, which have a slow rate of progression. The rod-derived cone viability factor (RdCVF), differences in macular pigment and peak distribution, cone density, greater vulnerability of some parafoveal photoreceptors, and variables associated with RPE and the choroid have all been brought up as potential underlying causes [2]. 

Some retinal/choroidal dystrophies are monogenic or digenic; however, in some cases, mutations in the same gene could lead to different phenotypes. To date, mutations in over 270 genes have been associated with retinal/choroidal dystrophies. However, molecular diagnosis still remains elusive in about a third of cases due to extreme genetic heterogeneity [2,3]. Various publications use the term atrophy rather than dystrophy since it is frequently impossible to establish an underlying genetic pattern. It was Knapp who reportedly coined the term “circinate choroidal sclerosis” in 1907 in order to describe paramacular choriocapillaris atrophy, while Sorby used the term “paracentral choroidal sclerosis” to describe a case that was comparable in 1938. Both Schocket and Ballin (1970) and Krill and Archer (1971) describe later cases [4,5]. Assuming an autosomal dominant and autosomal recessive pattern of inheritance, the genetic pattern is unknown. The disease is extremely uncommon; just a few cases have been reported, and OMIM does not even include it [5].

The objective of this study was to present a case of paramacular choriocapillaris atrophy with a foveal-sparing phenotype and to conduct multimodal imaging.

## 2. Case Report

The 73-year-old patient that stated they had impaired vision and photophobia in both eyes during a regular ophthalmological examination denied having visual field defects and night blindness. According to the patient’s medical history, she has been treated for diabetes type II and arterial hypertension for the last 5 years. Her deceased sister was legally blind at the age of 60. Best-corrected visual acuity measurements on ETDRS charts, applanation tonometry, and examination of the anterior and posterior segment were performed. Further tests were carried out, including fundus autofluorescence (FAF), fluorescein angiography (FA), optical coherence tomography with angiography (OCT/OCTA), computerized perimetry, and electroretinography (ERG). Immunological and infectious histories were both negative. 

## 3. Results

The right eye BCVA was 0.7, and the left eye BCVA was 0.6. Anterior biomicroscopy detected incipient corticonuclear cataracts. The intraocular pressure was normal. An examination of the fundus in mydriasis revealed an optic disc with clear margins, a fovea with spared structure, and a bilateral, concentric zone of grayish atrophy within the temporal arcades, along with visible choroidal vessels (Figure 1A,B). Retinal vessel attenuation, pale papillae, and bone spicule pigmentation were not detected. 

On the FAF, a zone of hypoautofluorescence corresponding to the atrophic area was observed, while the fovea showed hyperautofluorescence in both eyes. Numerous hyper/hypoautofluorescences were seen at the boundary of the appearance-preserved and atrophic chorioretina, with the optic disc and upper temporal arcade showing the greatest intensity (Figure 2A,B). In the early phase of FA, the lack of filling of the retinal vasculature and choriocapillaris caused a hypofluorescent zone, whereas the fovea and the margins of the atrophic area were surrounded by hyperfluorescence, which in the later phase showed signs of leakage from preserved choriocapillaris. A “window defect” was observed in the peripapillary area and along the lesion’s borders, particularly in the region of the upper temporal arcade, which corresponds to the area of RPE degeneration and the still-preserved choriocapillary layer (Figure 3A,B).

The OCT performed revealed the normal range of RNFL thickness by age (Figure 4). Furthermore, the OCT analyzed the parafoveal atrophy of the outer layers of the retina as well as sparing the external limiting membrane (ELM) ellipsoid zone and RPE in the fovea. Circular intraretinal edema was detected at the boundary of the fovea and parafovea (Figure 5A,B). The OCTA showed a reduced retinal capillary plexus and choriocapillary layer (Figure 6A–F). The visual field suggested the presence of relative and absolute central and paracentral scotomas (Figure 7A,B). Although the fovea appeared to be intact, diffuse photoreceptor dysfunction as well as reduced oscillatory potentials with delayed implicit time were shown on the ERG.

Following the ophthalmological clinical examinations and tests, genetic testing was performed at the BluePrints Genetic Laboratory in Finland. A panel of 314 genes for retinal dystrophies as well as 37 mitochondrial genes were studied. However, the likely genetic heritage has remained unknown. The patient is heterozygous for BEST1 c.1681A>G, p.(Thr561Ala), RP1L1 c.2107C>G, p.(Arg703Gly), and TTLL5 c.3425C>T, p.(Thr1142Ile), which are variants of uncertain significance. 

All ophthalmological clinical findings and diagnostic tests performed with results are summarized in Table 1 and Table 2.

## 4. Discussion

After carrying out the literature search, we considered regional choroidal dystrophies such as central areolar choroidal dystrophy (CACD) and paramacular choriocapillaris atrophy, as well as Stargardt disease type I (STGD1) and pseudo-Stargardt pattern dystrophy (PSPD) with a foveal-sparing phenotype due to the patient’s posterior-pole-specific atrophy and clinical test results. In terms of the genetic risk factors, it has been suggested that the presence of relatively mild genetic variants is correlated with a less severe phenotype and a typically later age of onset. In a recent study, a combination of a severe and a mild ABCA4 variation or only one ABCA4 variant was identified in the majority of STGD1 patients with foveal sparing [6]. PSPD with foveal sparing was identified as cases with a clinical Stargardt-like phenotype and a mutation in the PRPH2 gene, with or without an additional variance in a single ABCA4 allele, while a typical CACD phenotype and a single PRPH2 gene variant were thought to be enough to make the diagnosis of CACD [7]. In our study, the underlying genetic pattern remains unclear, which points to paramacular choriocapillaris atrophy.

Although atrophy of the choriocapillary layer is considered to be the crucial event in the regional choroidal dystrophies in the early described cases, early ophthalmoscopic signs of the disease at the level of the RPE, such as patches of hyperpigmentation and depigmentation, were described even back then [4,5]. The presence of hyperautofluorescence in the fovea on FAF images may indicate RPE activity, whereas several indicators of hyper/hypoautofluorescence along the margin of the atrophic and seemingly intact retina may indicate disease progression. The RPE is degenerating in the peripapillary area and in the area of the upper temporal arcade in the FA, while the choriocapillary layer appears intact. This supports recent histology research that choroidal dystrophies and Stargardt disease are primarily caused by alterations at the level of the retinal pigment epithelium (RPE) [1]. Choriocapillaris atrophy, visibility of choroidal blood vessels, and lesion expansion are all seen in the later phases of paramacular choriocapillaris atrophy [8].

All of the conditions above appear to have preserved the fovea at the onset of the disease. Frequently, the parafoveal retina is where patches of chorioretinal atrophy first appear. Multifocal atrophic zones can consolidate over time as a result of dissemination, and new atrophic areas can develop as well. The fovea might seem unaffected by chorioretinal atrophy on clinical examination until the late stage of disease [9,10]. As delayed foveal degeneration lengthens the window for using therapeutic approaches like gene therapy, patients with foveal sparing may become candidates for upcoming therapeutic studies [11]. There have been cases reported in the literature of a spared fovea in patients with paramacular choriocapillary atrophy, even after 20 years of patient follow-up [8]. Circular intraretinal edema in SD-OCT images that were detected along the boundary of the spared fovea and zone of parafoveal atrophy might be indicative of exudative maculopathy. It also might be indicative of disease progression towards the fovea. It is thought that central areolar choroidal dystrophy originated as an edematous/exudative maculopathy [12]. The modifications of widespread photoreceptor malfunction revealed on the ERG are consistent with examples published in the literature in which there is a decrease in cones and subclinical involvement of the central macula despite an apparently spared fovea [8,13].

The fundamental mechanism of foveal sparing is not well understood. The significance of understanding the natural history of this peculiar phenomenon increases in light of recent therapeutic approaches currently being evaluated to prevent or slow the progression of chorioretinal atrophy in choroidal/retinal dystrophies (e.g., NCT01736592, NCT01469832, and NCT02402660 on www.clinicaltrials.gov, (accessed on 3 July 2023)). The similarity between the foveal sparing phenomenon in monogenic and multifactorial diseases gives support to the idea that disease-independent mechanisms are responsible for shaping the foveal-sparing phenotype [2,14]. It has been suggested that the specific choroidal blood supply to the fovea is a contributing component to the local protective effect [15]. Furthermore, a substance known as the RdCVF protects the cone cells from degeneration. It is possible that higher levels of RdCVF secretion combined with increased foveal cone sensitivity will improve central cone survival [16]. Another explanation could be found in the peak cone density, which varies greatly and ranges from 98,200 to 324,100 cones/mm^2^. The phenomenon of foveal sparing may be influenced by this exceptional interindividual variability, which is much less prominent in the area around the fovea [17]. Furthermore, the foveal center lacks rods and S-cones, and the relative preservation of the fovea in some patients may be explained by their increased susceptibility to disease and/or aging. The unfavorably high ratio of rods to RPE cells in the parafoveal area, which can lead to an earlier decompensation of metabolic function and accelerate perifoveal atrophy, may also be a contributing factor [18,19]. The most central photoreceptors in eyes with foveal sparing may also be preserved as a result of an unequal distribution of macular pigment. This leaves the parafoveal photoreceptors comparatively vulnerable. By filtering potentially harmful blue light and acting as antioxidants, these carotenoids (lutein, zeaxanthin, and meso-zeaxanthin) protect against macular degeneration. Finally, genetic variances may influence the durability of the fovea [20].

It can be said that a new era has begun with the recent approval of Luxturna, the first gene therapy treatment for Leber congenital amaurosis and retinitis pigmentosa. Gene replacement strategies have so far produced the best outcomes in the treatment of retinal dystrophies [21]. A murine model of autosomal recessive RP, which is characterized by nonsense homozygous mutations in the PRPH2 gene, provided evidence of the therapeutic potential of gene replacement. The creation of the disc and restoration of the structural integrity of the photoreceptors were made possible by the subretinal injection of a viral vector bearing the PRPH2 transgene [22]. Another example of gene replacement therapy could be the injection of the viral vector carrying ABCA4, and as a result, the retina’s morphology and function might improve as a consequence of the decreased lipofuscin level [23]. The methodological advancements in genome sequencing technologies that we have seen over the past ten years have marked a turning point in the identification of new therapeutic approaches as well as in diagnosis and prognosis. Finally, the long-term preservation of the fovea is essential for daily life and, consequently, for quality of life [14].

Further objectives include long-term patient follow-up, which may aid in understanding the progression of the disease. The visual acuity tests, visual field evaluation, electroretinographic recordings, structural imaging FAF, SD-OCT, and OCTA can all be used to monitor patients who have a progressive visual impairment. The characteristics of the foveal sparing morphology and atrophy progression during the patient’s follow-up could be identified via the qualitative analysis of available FAF and SD-OCT images. A clearly defined zone of dramatically reduced autofluorescence can be associated with the atrophy of the retinal pigment epithelium and outer retina, while SD-OCT could be utilized to confirm the integrity of the foveal outer retina. The FAF intensity of the border region has been reported to be of prognostic value [24,25]. Therefore, the means to analyze FAF lifetimes in the border region are of great value and may be clinically useful in the future.

## Figures and Tables

**Figure 1 biomedicines-11-02074-f001:**
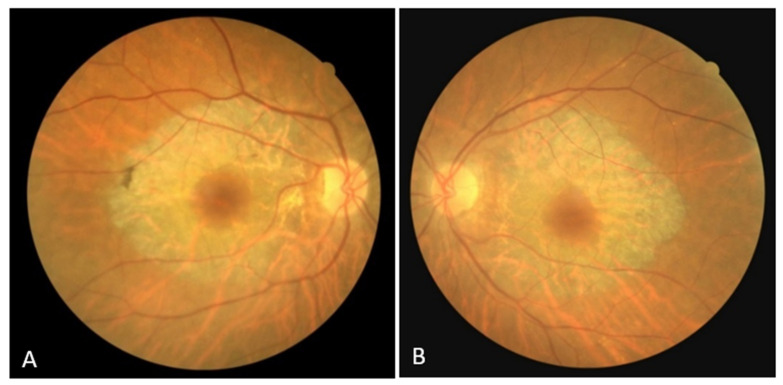
(**A**) Color fundus image of the right eye; (**B**) color fundus image of the left eye.

**Figure 2 biomedicines-11-02074-f002:**
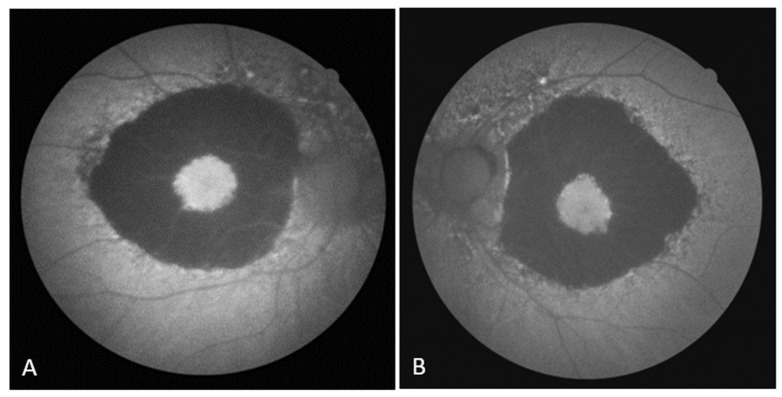
(**A**) FAF image of the right eye; (**B**) FAF image of the left eye.

**Figure 3 biomedicines-11-02074-f003:**
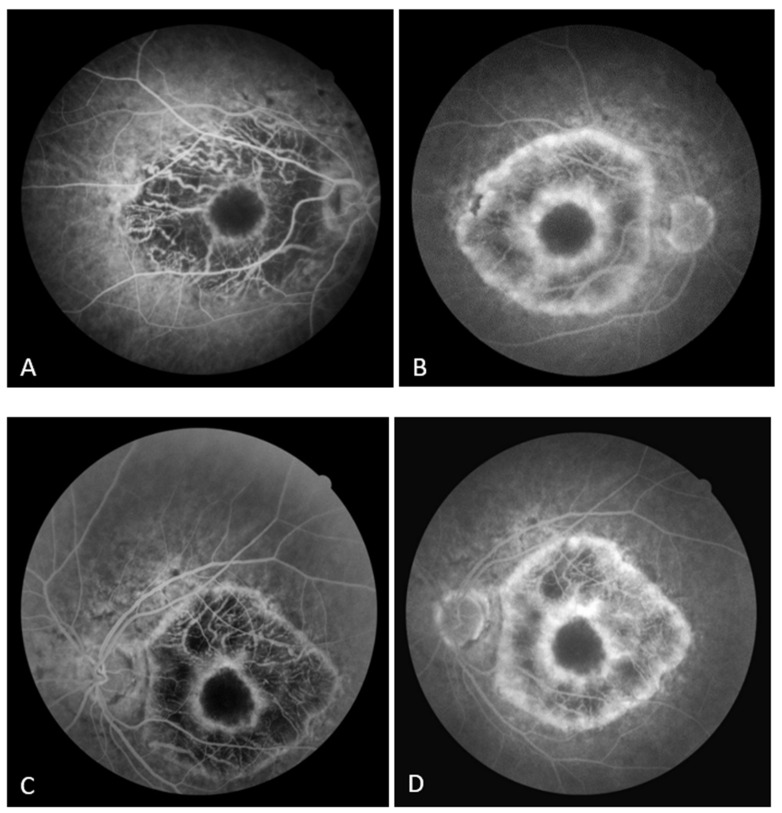
(**A**) Early FA image of the right eye; (**B**) late FA image of the right eye. (**C**) Early FA image of the left eye; (**D**) late FA image of the left eye.

**Figure 4 biomedicines-11-02074-f004:**
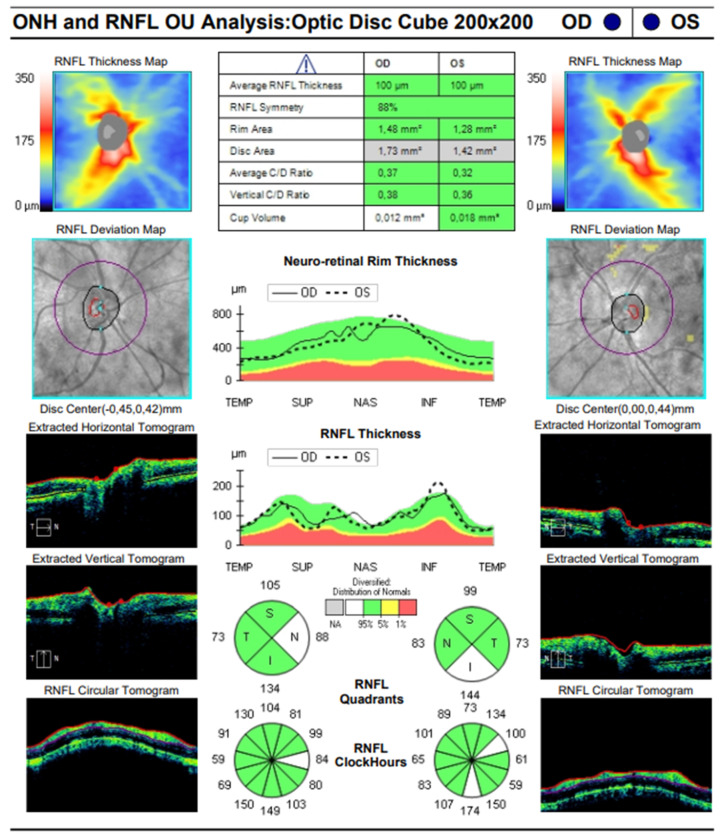
SD-OCT image of the optic disc of both eyes.

**Figure 5 biomedicines-11-02074-f005:**
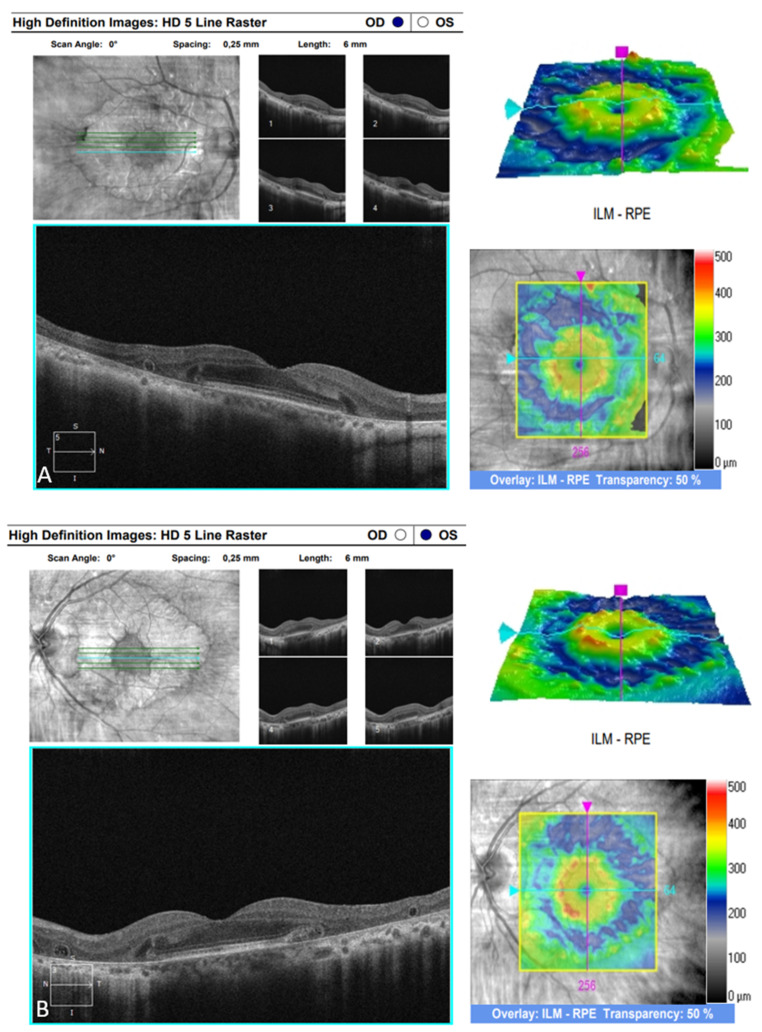
(**A**) SD-OCT image of the right eye. (**B**) SD-OCT image of the left eye.

**Figure 6 biomedicines-11-02074-f006:**
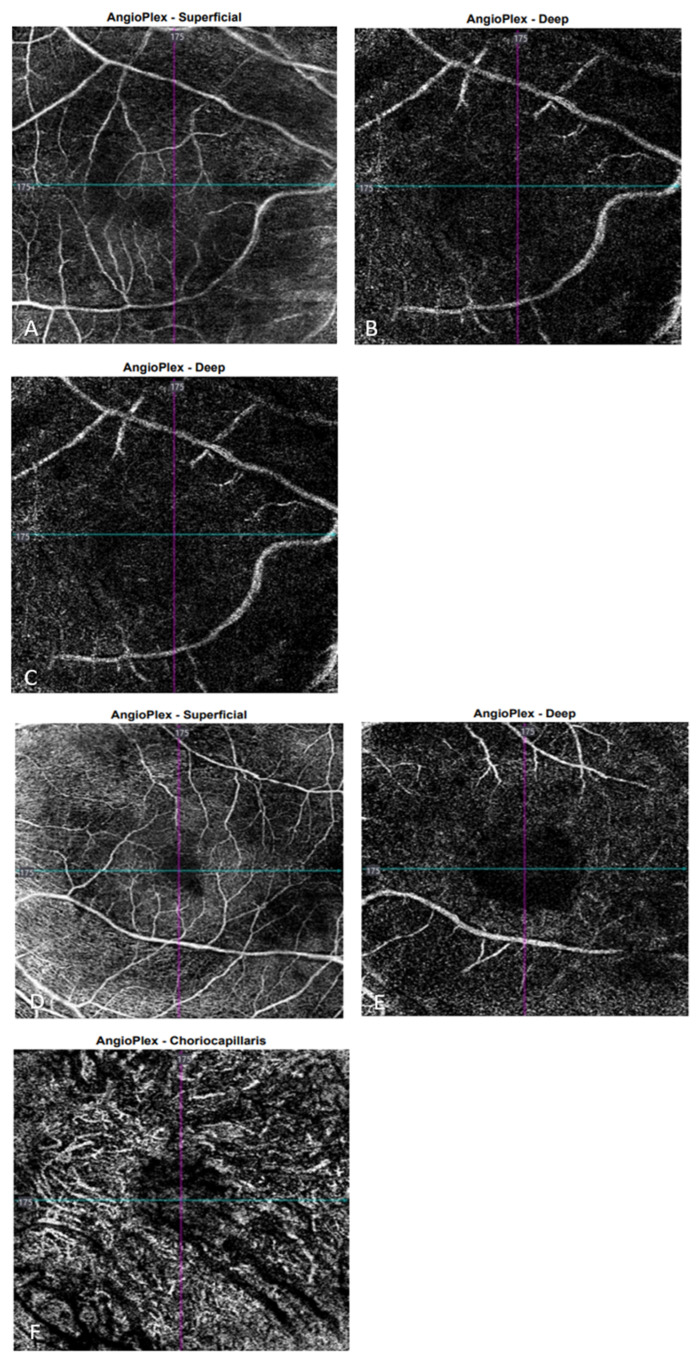
OCTA images of the right eye show the superficial capillary plexus (**A**), the deep capillary plexus (**B**), and choriocapillaris (**C**), respectively. OCTA images of the left eye show the superficial capillary plexus (**D**), the deep capillary plexus (**E**), and choriocapillaris (**F**), respectively.

**Figure 7 biomedicines-11-02074-f007:**
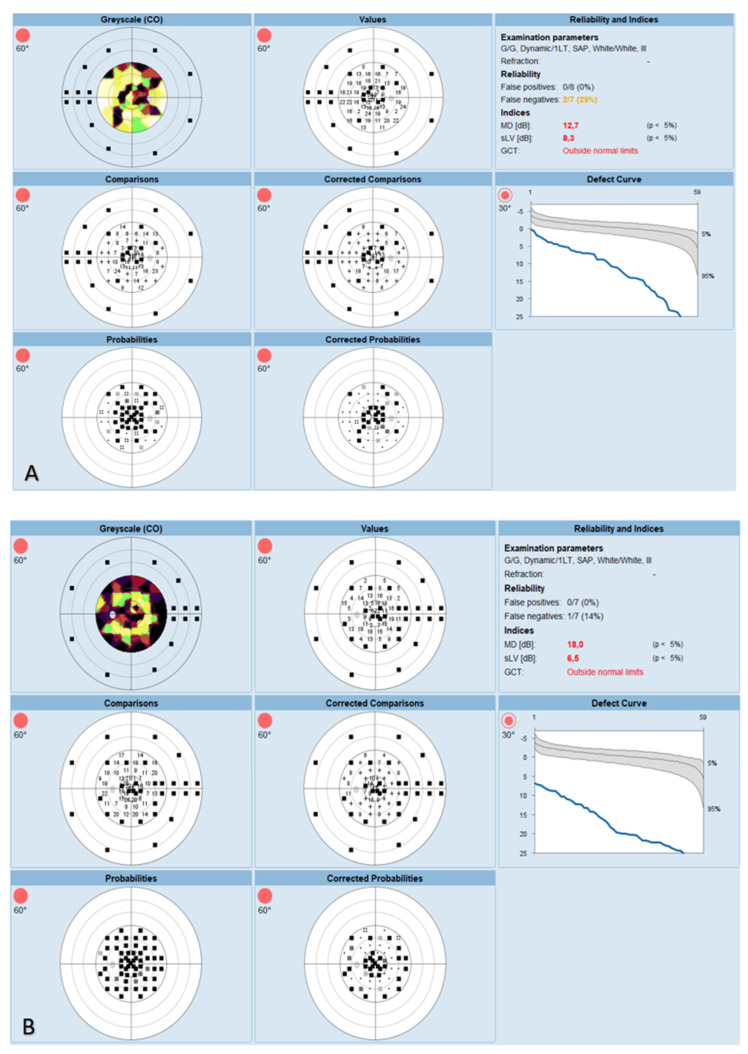
(**A**) Visual field of the right eye. (**B**) Visual field of the left eye.

**Table 1 biomedicines-11-02074-t001:** Ophthalmological clinical findings.

	Right Eye	Left Eye
Best-corrected visual acuity	0.7	0.6
Intraocular pressure (mmHg)	15	14
Anterior segment	no signs of inflammation;	no signs of inflammation;
pupils	round, correctly positioned,	round, correctly positioned,
	normal pupil response;	normal pupil response;
lens (LOCS II)	NI, CII, and PI	NI, CII, and PI
Fundus		
optic disc	clear margins;	clear margins;
	no signs of pallor (Figure 1A)	no signs of pallor (Figure 1B)
macula	fovea with spared structure	fovea with spared structure;
	symmetrical, concentric zone	symmetrical, concentric zone
	of grayish atrophy within the	of grayish atrophy within the
	temporal arcades with	temporal arcades with
	visible choroidal vessels	visible choroidal vessels
	(Figure 1A)	(Figure 1B)
vessels	no signs of attenuation	no signs of attenuation
		(Figure 1B)
	paravascular accumulation of	
	pigment close to the boundary	
	of the appearance preserved	
	and atrophic chorioretina	
	(Figure 1A)	
periphery	normal; no signs of bone cells	normal; no signs of bone cells
	or retinal degeneration	or retinal degeneration

**Table 2 biomedicines-11-02074-t002:** Ophthalmological diagnostic tests.

	Right Eye	Left Eye
Fundus autofluorescence		
(Canon CX-1)	hyperautofluorescence of the fovea;	symmetrical findings (Figure 2B)
	zone of the outer hypoautofluorescence	
	corresponding to the atrophic macular area;	
	numerous hyper/hypoautofluorescences	
	at the boundary of the appearance preserved	
	and atrophic chorioretina, with the optic disc	
	and upper temporal arcade showing the	
	greatest intensity (Figure 2A)	
Fluorescein angiography		
(Canon CX-1)	hypofluorescent zone due to the lack of filing of	symmetrical findings
early phase	the retinal microvasculature and choriocapillaris;	(Figure 3C,D)
	hyperfluorescence surrounding the fovea and the	
	margins of the atrophic area;	
	“window defect” in peripapillary area and in the	
	upper temporal arcade (Figure 3A)	
late phase		
	leakage from the preserved choriocapillaris	
	(Figure 3B)	
SD-OCT		
(CZM Cirrus HD-OCT)		
- optic disc	average RNFL thickness—normal (Figure 4)	average RNFL thickness—normal
		(Figure 4)
- macula	the ELM, ellipsoid zone, and RPE are still	symmetrical findings (Figure 5B)
	present in the fovea; visible disruption	
	of the outer layers with circular intraretinal	
	edema at the boundary of the fovea and	
	parafovea; atrophy of the outer layers in	
	parafoveal area; rosette-like structures;	
	visible Sattler’s and Haller’s layer	
	(Figure 5A)	
SD-OCTA		
	reduced retinal capillary plexus and	symmetrical findings (Figure 6D–F)
	choriocapillaris (Figure 6A–C)	
Computerized		
perimetry	relative and absolute central and paracentral	more affected;
(Octopus)	scotomas; MD 12.7 dB (Figure 7A)	MD 18.0 dB (Figure 7B)
Electroretinogram		
	diffuse photoreceptor reduction as well as	more pronounced
	reduced oscillatory potentials with delayed	
	implicit time	

## Data Availability

The data presented in this case report are available on request from the corresponding author.

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
