# Peer review of "Paramacular Choriocapillaris Atrophy"

_biomedicines, 2023, doi:10.3390/biomedicines11072074_

Round 1
Reviewer 1 Report
The paper entitled “Paramacular choriocapillaris atrophy” is based on an interesting case report regarding a rare case of paramacular choriocapillaris atrophy with foveal-sparing phenotype.
The results present the signs and clinical characteristics of this. Considering that this clinical presentation is rare, studies reporting specific cases with informative instrumental results scans can help in the clinical setting when faced with patients with untypical scans.
It would be interesting if the authors could further comment on differential diagnosis. Although the clinical signs and retinal characteristics have an unknown etiopathogenesis in this patient, additional information should be provided based on current literature regarding possible pathogenic pathways and mechanisms that could play a factor in this patient’s condition. Additional references based on similar cases need to be included. Mention should also be made about possible treatments that could prevent progression and further visual loss in this cases like this. These additional detail can help render the manuscript less superficial in nature and more clinically applicable. The study adds to the literature and is of potential interest, yet requires thorough modifications.
The table and figures are interesting and assist in describing the results. A flowchart on how to diagnose, manage and treat patients with atypical retinal signs and symptoms, however, could be considered to render this case report of clinical use.
Extensive editing by a native English doctor is needed to improve the English and flow of the text.
Author Response
Dear,
We would like to thank you for careful and thorough reading of this manuscript and for the thoughtful comments and constructive suggestions, which help to improve the quality of this manuscript. The corresponding changes and refinements made in the revised manuscript are summarized in our point-by-point responses below.
In introduction and in the first part of the discussion, we covered the genetic basis of differential diagnoses (line 39-43; 135-143). In the following, we explained in more detail the possible pathogenetic mechanisms of the foveal sparing (line 29-38; 173-197) and mentioned possible genetic treatments (line 198-211).
The manuscript was checked by M.Ed. in English language and literature.
We will be happy to make further adjustments, if necessary.
Sincerely, Ivona Bućan and Kajo Bućan
Reviewer 2 Report
hors:
The manuscript entitled “Paramacular choriocapillaris atrophy” describes a rare case of paramacular choriocapillaris atrophy with a foveal-sparing phenotype. In conclusion, the authors suggest that several heterogeneous dystrophies have phenotypically variable disease features with spared fovea and central visual acuity. This supports a disease-independent mechanism that allows the survival of foveal cones. The current findings are interesting.
Comments:
1) In the introduction the authors should describe in more detail some of the proposed mechanisms for the etiopathogenesis of paramacular choriocapillaris atrophy.
2) To clarify it for young researchers, the authors are encouraged to elaborate in the manuscript about the main advantages of the used diagnostic tests, including fundus autofluorescence, fluorescein angiography, optical coherence tomography with angiography, computerized perimetry, and electroretinography.
3) The discussion contains interesting points. However, the authors should put more effort into integrating/replacing their findings in the context of current knowledge. From a mechanistic perspective, the authors are advised to elaborate on how the genetic makeup is involved in the etiopathogenesis of paramacular choriocapillaris atrophy.
4) The authors should discuss the potential clinical significance of the results obtained.
5) More recent references are advised to be added to the current manuscript.
6) The manuscript needs to be revised carefully by a native English speaker. For example:
A) In line 9, the authors state “The 73-year-old patient stated impaired vision and photophobia on both eyes”. Please, replace “on” by “in”.
B) In lines 10-11, The authors state “Complete opthalmological examination and diagnostic tests, including …”. Please, replace “opthalmological” by “ophthalmological”.
Moderate editing of the English language is recommended.
Author Response
Dear,
We would like to thank you for careful and thorough reading of this manuscript and for the thoughtful comments and constructive suggestions, which help to improve the quality of this manuscript. The corresponding changes and refinements made in the revised manuscript are summarized in our point-by-point responses below.
1) In the introduction and in the discussion, we explained in more detail the assumed etiopathogenetic mechanisms (line 29-38; line 173-197).
2) We have written all the results obtained from the diagnostic tests in the manuscript text (line 72-90). Also in the discussion part, we presented two important features of disease progression monitoring using FAF, FA and SD-OCT images (line 148-154; 165-169). Finally, we mentioned the SD-OCT and FAF images as important tool that can be used to monitor patients (line 213-222).
3)The genetic background of paramacular choriocapillary atrophy remains unknown. In the discussion, a possible genetic basis is added, which is assumed to be significant in retinal/choroidal dystrophies with a preserved fovea (line 39-43; 135-143).
4) The clinical significance of the presented case consists in the further monitoring of the patient through OCT and FAF (line 212-222).
5) We have added new references.
6) The manuscript was checked by M.Ed. in English language and literature.
We will be happy to make further adjustments, if necessary.
Sincerely, Ivona Bućan and Kajo Bućan
Round 2
Reviewer 1 Report
The authors have addressed the issue raised in a sufficient manner.
Minor English revisions are still needed to enhance the flow of the paper.
Author Response
Dear reviewer,
Regarding the English language, the text has passed the revision (MDPI English-Editing-Certificate-68960).
We will be happy to make further adjustments, if necessary.
Sincerely, Ivona Bućan and Kajo Bućan